# Quantitative Proteomics Reveals That ADAM15 Can Have Proteolytic-Independent Functions in the Steady State

**DOI:** 10.3390/membranes12060578

**Published:** 2022-05-31

**Authors:** Chun-Yao Yang, Simone Bonelli, Matteo Calligaris, Anna Paola Carreca, Stephan A. Müller, Stefan F. Lichtenthaler, Linda Troeberg, Simone D. Scilabra

**Affiliations:** 1Centre for Osteoarthritis Pathogenesis Versus Arthritis, Kennedy Institute of Rheumatology, University of Oxford, Roosevelt Drive, Oxford OX3 7FY, UK; gonadotropin1703@gmail.com; 2Proteomics Group of Fondazione Ri.MED, Research Department IRCCS ISMETT (Istituto Mediterraneo per i Trapianti e Terapie ad Alta Specializzazione), Via E. Tricomi 5, 90145 Palermo, Italy; sbonelli@fondazionerimed.com (S.B.); mcalligaris@fondazionerimed.com (M.C.); apcarreca@fondazionerimed.com (A.P.C.); 3Department of Pharmacy, University of Pisa, Via Bonanno 6, 56126 Pisa, Italy; 4German Center for Neurodegenerative Diseases (DZNE), Feodor-Lynen Strasse 17, 81377 Munich, Germany; stephan.mueller@dzne.de (S.A.M.); stefan.lichtenthaler@dzne.de (S.F.L.); 5Neuroproteomics, School of Medicine, Klinikum Rechts der Isar, Technical University of Munich, 81675 Munich, Germany; 6Munich Cluster for Systems Neurology (SyNergy), 81377 Munich, Germany; 7Norwich Medical School, University of East Anglia, Bob Champion Research and Education Building, Rosalind Franklin Road, Norwich NR4 7UQ, UK

**Keywords:** metalloproteinases, ADAMs, ADAM15, proteomics, secretome

## Abstract

A disintegrin and metalloproteinase 15 (ADAM15) is a member of the ADAM family of sheddases. Its genetic ablation in mice suggests that ADAM15 plays an important role in a wide variety of biological functions, including cartilage homeostasis. Nevertheless, while the substrate repertoire of other members of the ADAM family, including ADAM10 and ADAM17, is largely established, little is known about the substrates of ADAM15 and how it exerts its biological functions. Herein, we used unbiased proteomics to identify ADAM15 substrates and proteins regulated by the proteinase in chondrocyte-like HTB94 cells. ADAM15 silencing did not induce major changes in the secretome composition of HTB94 cells, as revealed by two different proteomic approaches. Conversely, overexpression of ADAM15 remodeled the secretome, with levels of several secreted proteins being altered compared to GFP-overexpressing controls. However, the analysis did not identify potential substrates of the sheddase, i.e., transmembrane proteins released by ADAM15 in the extracellular milieu. Intriguingly, secretome analysis and immunoblotting demonstrated that ADAM15 overexpression increased secreted levels of tissue inhibitor of metalloproteinases 3 (TIMP-3), a major regulator of extracellular matrix turnover. An inactive form of ADAM15 led to a similar increase in the inhibitor, indicating that ADAM15 regulates TIMP-3 secretion by an unknown mechanism independent of its catalytic activity. In conclusion, high-resolution quantitative proteomics of HTB94 cells manipulated to have increased or decreased ADAM15 expression did not identify canonical substrates of the proteinase in the steady state, but it revealed that ADAM15 can modulate the secretome in a catalytically-independent manner.

## 1. Introduction

A disintegrin and metalloproteinase 15 (ADAM15) is a member of the ADAM family of metalloproteinases that plays important roles in a wide variety of biological functions, including cartilage homeostasis [1,2]. High levels of ADAM15 are expressed in vascular cells, endocardium, hypertrophic cells in developing bone, and specific areas of the brain, however, *Adam15*-null mice do not show any obvious developmental defects [1]. Conversely, *Adam15*-null mice developed accelerated age-dependent osteoarthritis (OA), suggesting that ADAM15 has protective effects on the joint. As a consequence of its ability to regulate cell adhesion and migration, ADAM15 is also associated with various inflammatory diseases including atherosclerosis [3,4], lung inflammation [4,5], rheumatoid arthritis [6], and inflammatory bowel disease [7]. Despite these potentially important roles, little is known about how ADAM15 exerts its biological functions in these processes and which proteins are cleaved or shed by the proteinase. A number of ADAM15 substrates have been identified using in vitro screening systems. For example, CD23 was shown to be cleaved by ADAM15 in a peptide library screening experiment [8]. Similarly, FGFR2iiib was identified as an ADAM15 substrate using alkaline phosphatase (AP)-tagging of candidate substrates in cell-based assays [9]. Additionally, ADAM15 has been reported to activate pro-MMP-9 and enhance MMP-9-mediated gelatin degradation [10]. However, these proteins have not yet been validated as physiological ADAM15 substrates, and the functional relevance of their ADAM15-mediate processing in vivo is unknown. Moreover, the number of ADAM15 substrates that have been identified so far is fairly limited compared to the substrate repertoire of other members of the family, including ADAM10 and ADAM17 [11,12]. Thus, we used quantitative high-resolution proteomics in order to uncover novel substrates of ADAM15 which could link its activity with the biological processes in which ADAM15 is known to be involved, for example cartilage homeostasis and cell adhesion/migration. We found that silencing of ADAM15 in chondrocyte-like cells did not induce significant alterations in their secretome. Similarly, ADAM15 overexpression did not increase levels of released transmembrane proteins in these cells. Intriguingly, we found that levels of tissue inhibitor of metalloproteinases 3 (TIMP-3), a secreted protein that prevents cartilage catabolism, was augmented in ADAM15-overexpressing cells. Similar changes in TIMP-3 levels were seen with an inactive form of ADAM15, in which the catalytic glutamate was mutated to alanine (ADAM15 E349A) indicating that ADAM15 regulation of TIMP-3 is independent of its catalytic activity. This study thus indicates that ADAM15 can alter levels of some soluble chondroprotective proteins (e.g., TIMP-3) independent of its catalytic activity. This provides new insights into the biology of ADAM15 and its involvement in cartilage homeostasis.

## 2. Materials and Methods

### 2.1. Cell Culture Reagents

Dulbecco’s modified Eagle’s medium (DMEM), DMEM without phenol red, L-glutamine, penicillin/streptomycin, and trypsin-EDTA were from Lonza (Basel, Switzerland); fetal calf serum (FCS) and phosphate buffered saline (PBS) from Gibco (part of Thermo Fisher Scientific, Waltham, MA, USA) HTB94 human chondrosarcoma cell line was purchased from American Type Culture Collection (ATCC), and maintained in DMEM with 10% FCS, 100 units/mL penicillin, and 100 units/mL streptomycin at 37 °C in 5% CO_2_.

### 2.2. Materials

DNA extraction was performed using QIAGEN Plasmid Plus Maxi Kits (QIAGEN, Cat. 12963, Hilden, Germany), RNA extraction was conducted using RNeasy Mini Kits (QIAGEN, Cat. 74104, Hilden, Germany), reverse-transcription was performed using High-Capacity cDNA Reverse Transcription Kits (Thermo Fisher Scientific, Cat. 4368814, Waltham, MA, USA) and quantitative PCR (qPCR) was performed using TaqMan Fast Universal PCR Master Mix (Thermo Fisher Scientific, Cat. 4352042, Waltham, MA, USA) with TaqMan Gene Expression Assays primers (Thermo Fisher Scientific, Waltham, MA, USA) for human ADAM15 (Hs00187052_m1), AXL (Hs01064444_m1), and RPLP0 (Hs99999902_m1).

### 2.3. Secretome Analysis of ADAM15-Silenced HTB94 Cells

HTB94 cells were plated in a 6-well plate the night before transfection. After washing, 1.8 mL of serum-free DMEM was added to each well and transient transfection was performed using jetPRIME transfection reagent (Polyplus, Illkirch-Graffenstaden, France) according to the manufacturer’s guide. Briefly, 200 μL jetPRIME Buffer was mixed thoroughly with 10 nM of siADAM15 (Thermo Fisher Scientific, Cat. s16681, Waltham, MA, USA), or the same concentration of the non-targeting control siRNA (Cat. 4390844) by vortexing. 4 μL of jetPRIME reagent was added to the jetPRIME Buffer-siRNA mixture and mixed thoroughly, incubated for 10 min at room temperature and added to wells. Cells were cultured at 37 ℃ for 24 h, changed to fresh serum-free DMEM without phenol red, and left for another 48 h for proteins to accumulate in the CM. CM was then collected in a 15 mL tube, and centrifuged (5 min, 510× *g*) to pellet cell debris. Conditioned media (CM) from all 6 wells in a 6 well plate were pooled together (12 mL total) and treated as one sample. 6 replicates of siADAM15 and 6 replicates of non-targeting control were analysed. Supernatants (12 mL) were collected, filtered through a 0.2 μm Corning syringe filter, and concentrated 50-fold (10 mL to 200 μL) on 10 kDa MWCO Microcon centrifugal filters (Millipore, part of Sigma Aldrich, St. Louis, MO, USA).

### 2.4. Sample Preparation for LC-MS/MS

A protein amount of 25 μg per sample was subjected to proteolytic digestion using the filter-assisted sample preparation (FASP) protocol with 30 kDa Vivacon spin filters (Sartorius, Göttingen, Germany) [13]. Proteolytic peptides were desalted by stop and go extraction (STAGE) with C18 tips [14]. The purified peptides were dried by vacuum centrifugation and then dissolved in 40 μL 0.1% formic acid for LC-MS/MS analysis.

### 2.5. LC-MS/MS Analysis

A peptide amount of 1 μg was separated on a nanoLC system (EASY-nLC 1000, Proxeon–part of Thermo Scientific, Waltham, MA, USA) using an in-house packed C18 column (50 cm × 75 μm ID, ReproSil-Pur 120 C18-AQ, 1.9 μm, Dr. Maisch GmbH, Germany) with a binary gradient of water (A) and acetonitrile (B) containing 0.1% formic acid (0 min, 2% B; 3:30 min, 5% B; 137:30 min, 25% B; 168:30 min, 35% B; 182:30 min, 60% B) at 50 °C column temperature. The nanoLC was coupled online via a nanospray ex ion source (Proxeon–part of Thermo Scientific, Waltham, MA, USA) to a Q-Exactive mass spectrometer (Thermo Scientific, Waltham, MA, USA). Full MS spectra were acquired at a resolution of 70,000. The top 10 peptide ions exceeding an intensity of 2.5 × 10^4^ were chosen for collision-induced dissociation. Fragment ion spectra were acquired at a resolution of 17,500. A dynamic exclusion of 120 s was used for peptide fragmentation.

### 2.6. “Label-Free Quantification” and Data Analysis

The data were analyzed by the software Maxquant (maxquant.org, Max-Planck Institute Munich) version 1.5.3.12, as previously described [15]. In brief, the MS data were searched against a reviewed canonical fasta database of Homo sapiens from UniProt (download: 2016-01-26; 20,192 entries). Label-free quantification (LFQ) of proteins required at least two ratio counts of unique peptides. Only unique peptides were used for quantification. The LFQ values were log2 transformed and statistically analysed by using the Perseus software (version 1.6.15.0; Max Planck Institute Munich, Germany) [16]. A two-sided homoscedastic student’s *t*-test was used to evaluate significantly regulated proteins between ADAM15 KD HTB94 or mock-transfected control cells. A false discovery rate of 0.05 and 0 to 0.1 was set as the threshold for statistically significant alterations.

### 2.7. hiSPECS Analysis of ADAM15-Silenced HTB94 Cell Secretome

ADAM15 was silenced in HTB94 cells as described in Section 2.3. After transfection with ADAM15 siRNAs or non-targeting siRNAs, cells were cultured at 37 ℃ for 12 h before the addition of 50 μM ManNAz sugar to the culture for another 24 h. Conditioned media were processed as described [17], filtered through a 0.2 μm, and incubated with concavalin A-conjugated magnetic beads (2 h, 4 °C). Magnetic beads were collected using a magnet and the supernatant was discarded. The beads were washed 3 times using a wash buffer (50 mM Tris-HCL, pH 7.5, 150 mM NaCl, 2 mM EDTA, 1% Triton), and proteins eluted using concanavalin A elution buffer (500 mM methylalpha-D-mannopyranoside, 10 mM EDTA, 20 mM Tris-HCl, pH 7.5). Eluted proteins were incubated with magnetic DBCO beads in the dark (16 h, 4 °C). Magnetic DBCO beads were collected using a magnet and the supernatant was discarded. DBCO beads were washed three times in MS SDS buffer (100 mM Tris-HCl, pH 8.5, 1% SDS, 250 mM NaCl), three times in MS urea buffer (8 M urea in 100 mM Tris-HCl, pH 8.5), and three times in 20% acetonitrile with 0.1% sodium deoxycholate. Proteins bound to the DBCO beads were digested with LysC for 3 h, followed by digestion with trypsin for 16 h. The resultant peptide fragments were reconstituted in 8% formic acid (FA) and desalted using the STAGE procedure before being analysed by LC-MS/MS and LFQ, as described in Section 2.5 and Section 2.6 [14].

### 2.8. Secretome Analysis of ADAM15-Overexpressing HTB94 Cells

The full-length human ADAM15 sequence (from the signal peptide and pro-domain to the intracellular domain, transcript variant 1, NM_207191.1, National Center for Biotechnology Information) with a C-terminal FLAG tag in a pCMV2 vector was purchased from Stratech Scientific Ltd. (Ely, UK). A pCMV2-GFP vector was purchased from Stratech Scientific Ltd. (Ely, UK). HTB94 cells were seeded (5 × 10^6^ cells in 10 mL full DMEM per 10 cm^2^ dish) the night before transfection. Cells were washed twice with 10 mL ice-cold PBS and once with 15 mL of ice-cold PBS immediately before transfection. Media were then changed to 10 mL Opti-MEM. Transient transfection was performed using TransIT 2020 transfection reagent, according to the manufacturer’s guide. Conditioned media from ADAM15-overexpressing HTB94 and GFP-overexpressing HTB94 were analysed. Cells were changed to fresh Opti-MEM (15 mL) 48 h after transfection and left for another 24 h for proteins to accumulate in the conditioned media. Conditioned media were then collected in a 50 mL tube, and centrifuged (5 min, 510× *g*) to pellet cell debris. The resultant supernatants were collected, filtered through a 0.2 mm Corning syringe filter, and concentrated 75-fold (15 mL to 200 µL) on 10 kDa MWCO Microcon centrifugal filters (Millipore, part of Sigma Aldrich, St. Louis, MO, USA), followed by a smaller capacity 5 kDa MWCO Vivaspin 500 (GE Healthcare, Chicago, IL, USA) spin filter. Conditioned media were then applied to FASP and STAGE tipping, tryptic peptides run into an LC/MS-MS and secretome analysed by label-free quantification, as described in Section 2.5 and Section 2.6.

### 2.9. Validation of Proteins by Immunoblotting

Conditioned media from ADAM15 KD and control HTB94 cells were collected and concentrated by TCA-precipitation. Similarly, conditioned media from cells overexpressing ADAM15, a mutated ADAM15(E349A) that was created by site-directed mutagenesis of glutamic acid residue 349 to alanine (E349A) using Quick Change II XL (Agilent Technologies, Stockport, United Kingdom), or control cells were collected and TCA-precipitated.

Cells were collected with STET lysis buffer (50 mM Tris, pH 7,5, 150 mM NaCl, 2 mM EDTA, 1% Triton), containing a protease inhibitor cocktail (1:500, Sigma, P-8340, Aldrich, St. Louis, MO, USA). Conditioned media and cell lysates were loaded onto an acrylamide gel and, after electrophoretic separation, blotted onto a PVDF membrane using the Trans-Blot Turbo transfer system (Biorad, Hercules, CA; USA) and detected by the following antibodies: anti-ADAM15 (clone EPR5619, Abcam, cat. ab124698), anti-FLAG M2 antibody (Sigma-Aldrich, cat F3165; Aldrich, St. Louis, MO, USA), AXL (R&D, cat AF154), TIMP-3 (Millipore, cat. AB6000), clusterin (clone B-5, Santa Cruz, cat. sc-6259), actin (clone D6A8, Cell Signaling, cat. 12620).

## 3. Results

### 3.1. Silencing of ADAM15 in Chondrocyte-Like Cells Had No Major Effects on Secretome Composition

ADAM15 has an active metalloproteinase catalytic motif and has been shown to cleave a number of proteins in vitro [9]. We used unbiased quantitative proteomics in order to identify new substrates of the proteinase. HTB94 cells were treated with ADAM15-targeting siRNAs to knock-down the expression of the proteinase. Expression of endogenous ADAM15 was reduced by more than 50% after 24 h and more than 80% reduction was achieved after 72 h (Figure 1A). Thus, after treatment with ADAM15- or non-targeting (NT) siRNAs, HTB94 cells were grown in the absence of serum for 72 h, and the conditioned media was analysed by LC-MS/MS and subsequent “label-free quantification” (LFQ). 2697 proteins were identified in the conditioned media of ADAM15 knocked-down (KD) cells or control cells. Among these, 642 were membrane proteins, comprising 115 type 1, 61 type 2, and 18 GPI-anchored proteins, and 227 were secreted proteins (according to Uniprot annotation; Figure 1B, Appendix A). We used LFQ analysis to detect alterations in protein levels in the secretome of ADAM15 KD cells and found that only one protein, the tyrosine-protein kinase receptor UFO (also known as AXL), was significantly reduced after the false discovery rate (FDR) correction for multiple hypotheses (and therefore above the FDR curves shown as black hyperbolic curves in Figure 1C; FDR *p* = 0.05; s0 = 0.1; Appendix A). AXL is a transmembrane type 1 protein known to be shed from the cell surface by ADAMs and is therefore a strong potential ADAM15 substrate [18]. To validate AXL as a novel ADAM15 substrate, we analysed its shedding using immunoblotting as an orthogonal method. Levels of shed AXL in the conditioned media of ADAM15 KD cells were reduced compared to controls, thus confirming the mass spectrometry-based LFQ results (Figure 1D,E). However, levels of full-length AXL in the cell lysate were also decreased, indicating that, rather than undergoing ectodomain shedding, the protein was regulated by a different mechanism (Figure 1D). We found that AXL mRNA expression was downregulated in ADAM15 siRNA-treated cells compared to controls (Figure 1F), suggesting that rather than being directly involved in the shedding of AXL, ADAM15 regulates levels of this protein through an unidentified transcriptional mechanism.

Serum starvation is commonly used in proteomics to evaluate levels of proteins in the conditioned media, as highly abundant serum proteins interfere with the detection of proteins that are truly secreted by cells in mass spectrometry-based experiments [19]. Nevertheless, the serum contains proteins, including growth factors, that are known to activate specific sheddases and thus promote the release of cell membrane proteins [20]. We reasoned that similar to ADAM17 [21], ADAM15 could be an activated sheddase that requires stimulation or activation in order to cleave and shed its substrates. Thus, we further analysed the secretome of ADAM15 KD HTB94 cells using “high-performance secretome protein enrichment with click sugars” (hiSPECS) [17]. This innovative method allows the isolation of proteins released by the cell using metabolic labelling followed by click-chemistry, thereby separating cell- and serum-derived proteins before mass spectrometry. hiSPECS identified 265 proteins in the conditioned media of ADAM15 KD or control cells, of which 102 were annotated as membrane and 129 as secreted proteins (Figure 1B,G, Appendix A). Secreted and shed proteins comprise about 85% of the total detected proteins, indicating a clear enrichment of these two groups of proteins by hiSPECS. Among the membrane proteins, 49 were type 1, 17 were type 2, 14 were GPI-anchored, and 22 were membrane proteins with a different topology (Figure 1B, Appendix A). Similar to the previous analysis, minimal changes were detected by hiSPECS in the secretome of ADAM15 KD cells (Figure 1G). Only one protein, dermicidin (DCD), was found to be significantly altered in this analysis. However, DCD, which is a secreted protein (according to its UniProt annotation), increased in the conditioned media of HTB94 cells upon ADAM15 silencing and therefore could not be a direct ADAM15 substrate, but rather is regulated by ADAM15 through a different unknown mechanism (Figure 1G, Appendix A). AXL, which was identified as the only hit by secretome analysis, was not detected by hiSPECS.

In conclusion, two separate proteomic methods showed that knockdown of ADAM15 led to minimal alterations in the secretome of chondrosarcoma-like cells.

### 3.2. Overexpression of ADAM15 Did Not Increase Release of Cell Membrane Proteins

Sheddases may have low constitutive activity, as has been demonstrated for ADAM17 [12]. Although we clearly detected ADAM15 in HTB94 cells by immunoblotting, we hypothesized that its levels were not sufficiently high to observe the consequences of its activity on ectodomain shedding. Thus, we overexpressed ADAM15 in HTB94 cells and used unbiased proteomics to evaluate its effects on protein shedding and to identify potential novel substrates of this metalloproteinase. HTB94 cells were transiently transfected with ADAM15 or GFP as a control, and ectopic expression was confirmed by immunoblotting (Figure 2A). Conditioned media were collected from transfected cells grown in the absence of serum for 48 h and analysed by mass spectrometry. Among the 1077 detected proteins in the conditioned media of ADAM15-overexpressing and GFP-transfected control HTB94 cells, 212 were annotated as membrane proteins (based on Uniprot annotation), of which 25 were type 1, 13 type 2, and 5 were GPI-anchored proteins (Figure 2B,C, Appendix A). In addition to membrane proteins, 121 secreted proteins were detected. LFQ analysis indicated levels of 261 proteins were altered above the FDR curves in ADAM15-overexpressing cells compared to controls (Figure 2D, Appendix A).

Among the altered proteins, 157 were more abundant in the conditioned media of ADAM15-overexpressing cells, while 103 proteins were less abundant. In order to analyse ectodomain shedding, we evaluated which of the proteins identified was annotated as being transmembrane in Uniprot. This showed that none of the identified transmembrane proteins were more abundant in the conditioned medium of ADAM15-overexpressing cells, indicating that the proteinase did not affect their release under these conditions.

However, 31 secreted proteins were identified as being altered in the conditioned media of ADAM15-overexpressing cells (Figure 2D, Appendix A), with 25 of them being less abundant. For example, levels of clusterin (CLU), a protein secreted via the conventional ER/Golgi pathway [22], were reduced in the media of ADAM15-overexpressing cells. 6 proteins were more abundant in the conditioned media of ADAM15-overexpressing cells, including PDC6I and CXL10. Intriguingly, TIMP-3 was the only protein found in all ADAM15-overexpressing cell replicates, while not detected in the GFP-expressing control samples (Figure 2E, Appendix A). Altogether, our results indicated while ADAM15 overexpression induced changes in the secretome, these were not directly related to its shedding potential.

### 3.3. ADAM15 Regulates Levels of TIMP-3

We found that TIMP-3, an endogenous inhibitor of cartilage degrading metalloproteinases such as matrix metalloproteinases (MMPs) and disintegrin metalloproteinases with thrombospondin domains (ADAMTSs) [23], was not detectable in the secretome of GFP-transfected cells, suggesting that its levels were below the detection threshold. TIMP-3 was however detectable in the secretome of cells overexpressing ADAM15, indicating ADAM15 increases its extracellular levels (Figure 2D,E, Appendix A). Immunoblotting analysis found a 50% increase in TIMP-3 in the conditioned media of ADAM15-overexpressing cells compared to controls (Figure 3A,B). Interestingly, overexpression of ADAM15 E349A (ADAM15 E/A), an inactive mutant of the proteinase [24], led to a similar increase in TIMP-3 in the conditioned media (Figure 3C,D). Clusterin was validated by immunoblotting as an example of a protein reduced by overexpression of ADAM15 (Figure 2D and Figure 3, Appendix A).

Altogether, the LFQ analysis found that overexpression of ADAM15 in chondrocyte-like HTB94 cells altered the abundance of several proteins in the conditioned media. None of these fit the criteria of a potential ADAM15 substrate (increased shedding of transmembrane proteins into conditioned media of ADAM15- but not ADAM15E/A-overexpressing cells), indicating that this membrane-tethered proteinase did not act as a sheddase under these conditions. ADAM15 did however increase levels of several soluble proteins including TIMP-3 in the conditioned medium in a catalytic activity-independent manner.

## 4. Discussion

ADAM15 belongs to the ADAM family of sheddases, a class of proteinases that are involved in the ectodomain shedding of transmembrane proteins and play a role in a number of biological processes including cell communication [25]. Both in vitro and in vivo studies have indicated a function for ADAM15 in a number of biological processes, including regulation of cartilage homeostasis. Nevertheless, its substrate repertoire and the molecular mechanisms by which it exerts this and other functions have not been established. Since other ADAMs, such as ADAM10, 12, and 17, have been found to principally shed proteins from the plasma membrane into the conditioned medium, and ADAM15 is known to be transported to the plasma membrane after maturation [26], we employed label-free quantitative proteomic methods optimised to detect changes in abundance of membrane and secreted proteins in the conditioned media upon knocking down ADAM15 expression in chondrocyte-like cells [17,19]. First, we analysed the secretome of ADAM15 KD cells using a quantitative, high-resolution mass spectrometric workflow that can be applied to cells grown in the absence of serum. This method allows the detection of low abundance proteins and proteins unconventionally secreted, including cytoplasmic and exosome-associated proteins [19]. Using this method, we identified one significant hit, AXL, which was decreased in the secretome of ADAM15-knockdown cells and was thus considered a potential substrate of ADAM15. AXL is a tyrosine kinase receptor belonging to the TAM family, and its interactions with its ligand, growth arrest-specific protein 6 (GAS6), are critical in controlling cell proliferation [27]. The ectodomain of AXL was significantly reduced in the secretome of ADAM15 KD cells, as we confirmed using immunoblotting as an orthogonal method, but a further analysis indicated that ADAM15 knockdown decreased mRNA and protein levels of AXL, indicating that ADAM15 regulates this protein through an undefined transcriptional mechanism rather than directly via ADAM15-mediated shedding. Since this approach for secretome analysis did not identify novel substrates of ADAM15, we performed another proteomic analysis using hiSPECS, which allows secretome analysis of cells cultured in the presence of serum [17], since serum and growth factors are known to stimulate shedding by ADAM proteinases [20,28]. We reasoned that serum may also stimulate ADAM15 activity, thus enabling the identification of its substrates. The hiSPECS method is based on the fact that most secreted proteins (66%) and potential shedding substrates (87% of type I and type II transmembrane proteins) are glycosylated as annotated in Uniprot. hiSPECS comprises metabolic labelling of cellular glycoproteins with azido sugars, and subsequent enrichment of these metabolically labelled glycoproteins from the conditioned medium by concanavalin A precipitation and coupling to magnetic dibenzylcyclooctyne (DBCO)–alkyne beads using copper-free click chemistry. This allows the separation of secreted and shed proteins from serum proteins and highly abundant cytosolic proteins released from apoptotic cells. hiSPECS successfully enriched secreted and shed proteins, since these accounted for more than 84% of all detected proteins. However, hiSPECS was unable to identify any novel protein candidates regulated by ADAM15, in agreement with a similar analysis of the *Adam15*-null chondrocyte secretome [29].

Several possibilities may explain why we were not able to identify the ADAM15 substrate using these hypothesis-free approaches. Firstly, quantitative secretome analysis relies on detecting proteins that are shed and released into the conditioned medium differentially when levels of a candidate sheddase are altered. Thus, this method cannot detect changes in the abundance of transmembrane proteins that are also cleaved by other proteinases which can compensate for reduced by ADAM15 (alternative processing). Additionally, this method is not suitable for detecting the shedding of transmembrane proteins that are then cleaved by other sheddases (sequential cleavage). Processing of APP is an example of both cases. Indeed, APP in neurons is mainly processed by ADAM10 and BACE1 under constitutive conditions [30,31], and, hence secretome analysis would hardly detect alternative cleavage of APP by a different sheddase if this occurs to a lesser extent. Furthermore, APP processing by membrane type-5 matrix metalloproteinase (MT5-MMP) is an example of the latter case [32]. MT5-MMP cleavage of APP occurs distantly from the cell membrane and releases a truncated ectodomain. After this shedding event, the truncated APP is further processed by ADAM10 and BACE1. Although this has functional consequences on neuronal activity in vivo, the MT5-MMP cleavage of APP could not be detected by hiSPECS and other methods for secretome analysis (unpublished observations). If ADAM15 acts in a similar manner to MT5-MMP, mediating alternative cleavages, its substrates will not be identified using such methods, which rely on protein abundance changes. Similarly, ADAM15 processing of ECM components or secreted proteins will also not be identified by such methods, since these proteins will be present in the conditioned medium regardless of ADAM15 activity. Another possibility is that ADAM15 functions as a non-sheddase, preferentially cleaving soluble proteins. For instance, both ADAM10 and ADAM17 can be shed themselves and, once in a soluble form, cleave secreted ECM components, such as PCOLCE and fibronectin [33]. ADAM15 may act in a similar manner. However, quantitative secretome analysis would also not detect changes in protein abundances in this case, and identification of soluble substrates would require other proteomic approaches, such as the Terminal Amine-based Isotope Labeling of Substrates (TAILS), which enables both substrate and cleavage site identification at the same time [34]. A third possibility is that ADAM15 has low constitutive activity and requires activation to exert substantial shedding activity. Notably, *Adam15*-null mice have relatively normal gross phenotypes, even in embryonic and adult tissues where *Adam15* expression is relatively high (e.g., hypertrophic chondrocytes, Purkinje cells and endothelial cells [1]), and abnormalities were only observed when animals were challenged (e.g., by reducing oxygen tension to induce neovascularisation [1]), or after prolonged periods of time (e.g., increased spontaneous OA after 12 months [2]). For this reason, we decided to overexpress ADAM15 in chondrocyte-like cells and test whether this approach could exhibit effects on the secretome. A number of membrane and secreted proteins were found to have significantly altered abundance in the secretome of ADAM15-overexpressing cells. Nevertheless, the analysis did not identify transmembrane proteins whose levels were increased in the conditioned media of ADAM15-overexpressing cells, and which are therefore potential ADAM15 substrates. ADAM15 overexpression did augment levels of several soluble proteins, including TIMP-3, in the extracellular milieu. Interestingly, overexpression of an inactive form of the proteinase promoted similar changes in TIMP-3, indicating that ADAM15 can play a role in the regulation of cartilage homeostasis by a mechanism that is independent of its catalytic activity. Although ADAM15 has a conserved metalloproteinase domain with a predicted active catalytic site, our results are in line with previous evidence that its primary functions are not related to shedding or protein cleavage. Indeed, Horiuchi et al. (2003) reported that murine ADAM15 is required for pathological neovascularisation but Maretzky et al. (2014) found that knock-in mice carrying a catalytically inactivate point mutation (E349A) showed comparable pathological neovascularisation to mice overexpressing wild-type ADAM15 [1,24]. This indicates that the catalytic function of ADAM15 is not required for its role in neovascularisation. Moreover, they were unable to identify any substrates relevant to angiogenesis, further supporting the possibility that ADAM15’s physiological and pathological roles may be independent of its shedding or proteolytic activity.

In conclusion, we used an array of proteomics methods to identify proteins regulated by ADAM15, whose substrate repertoire is still rather limited, with only a few proteins identified to be cleaved in vitro. Our study provides new insights into the biology of this proteinase and indicates that it can regulate the levels of other proteins involved in cartilage homeostasis in a catalytic-independent mechanism.

## Figures and Tables

**Figure 1 membranes-12-00578-f001:**
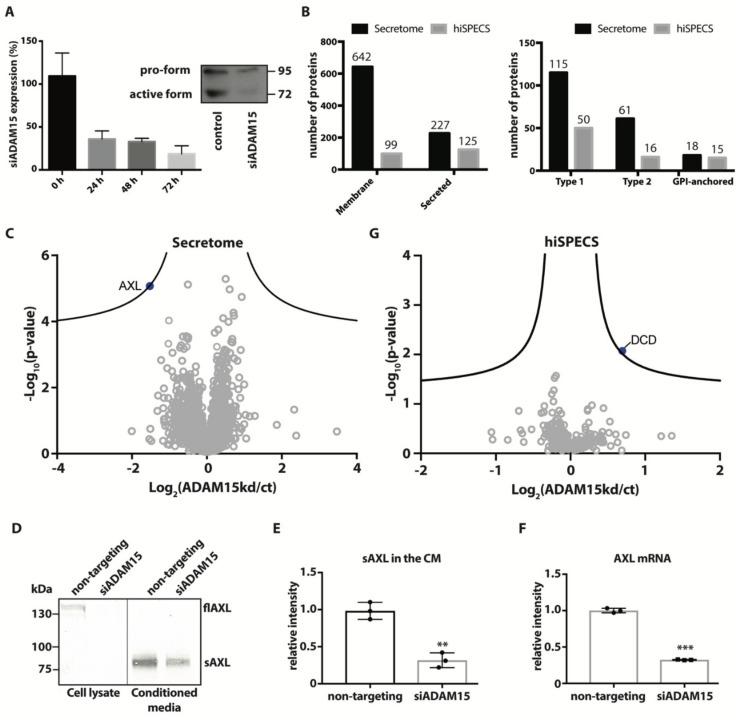
Secretome analysis of ADAM15 knockdown HTB84 cells. (**A**) HTB94 cells were transfected with 20 nM siRNA targeting ADAM15 (siADAM15) or same concentration of non-targeting control RNA (NT). ADAM15 expression was measured by qPCR at 0, 24, 48, and 72 h after treatment with ADAM15 or NT siRNAs, and displayed as a percentage of ADAM15 expression in cells transfected with NT siRNAs at each time point. Protein levels of ADAM15 in HTB94 cells treated with NT siRNAs (control) or ADAM15 siRNAs were analysed by immunoblotting at 72 h (insert). (**B**) Number of proteins annotated as a membrane or secreted proteins identified by secretome analysis or hiSPECS analysis in the conditioned media of ADAM15 knockdown HTB94 cells (left). Topology of single-span membrane proteins detected by secretome or SPECS analysis in the conditioned media of ADAM15 knockdown cells (right). (**C**) Volcano plot showing the −log_10_ of *p*-values versus the log_2_ of protein ratio between ADAM15 KD and HTB94 control cells (ct) of proteins detected in the conditioned media by serum-free secretome analysis (n = 6). AXL, which is the only protein above the FDR curves (displayed as black hyperbolic curves) is displayed as a blue solid dot, while proteins below the FDR are displayed by grey open dots. (**D**) Protein levels of full-length AXL in the cell lysates (flAXL) or shed AXL conditioned media (sAXL) of HTB94 cells treated with non-targeting or ADAM15 siRNAs (siADAM15) cells analysed by immunoblotting. (**E**) Bands corresponding to sAXL in the conditioned media from 3 independent experiments were quantified and normalized to the mean of sAXL values in conditioned media of cells treated with non-targeting siRNAs. A two-sided Student’s *t*-test was used to statistically evaluate changes in sAXL (** *p* < 0.01). (**F**) AXL mRNA expression levels in HTB94 cells treated with non-targeting or ADAM15 siRNAs plotted as relative 2^−ΔΔCT^ (*** *p* < 0.005, Student’s *t*-test) (**G**). Volcano plot showing the −log_10_ of *p*-values versus the log_2_ of protein ratio between ADAM15 KD and HTB94 control cells (ct) of proteins detected in the conditioned media by hiSPECS analysis (n = 6).

**Figure 2 membranes-12-00578-f002:**
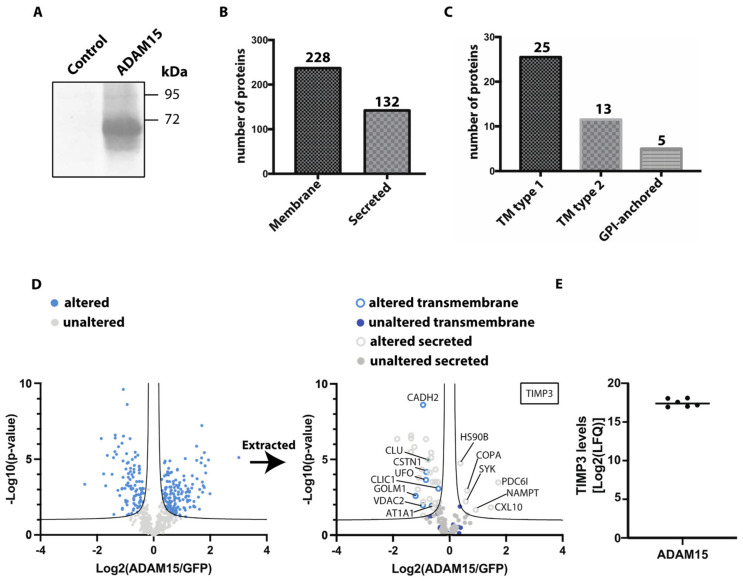
Secretome analysis of ADAM15-overexpressing HTB84 cells. (**A**) HTB94 cells were transiently transfected with ADAM15 or GFP as a control, and expression was evaluated by immunoblotting. (**B**) Number of proteins annotated as a membrane or secreted proteins detected in the secretome of ADAM15-overexpressing cells. (**C**) Topology of single-span membrane proteins detected in the secretome of ADAM15-overexpressing cells. (**D**) Volcano plot showing the −log_10_ of *p*-values versus the log_2_ of protein ratio between ADAM15-overexpressing and GFP transfected HTB94 control cells of 1077 proteins (n = 6). The hyperbolic curves indicate the false discovery rate. Proteins above these lines are considered significantly regulated, while proteins below are considered not regulated. TIMP-3 was found in all ADAM15-overexpressing replicates, while not being detected in GFP-expressing samples (insert) (**E**) Log2 of LFQ intensities for TIMP3 detected in ADAM15-overexpressing HTB94 cells.

**Figure 3 membranes-12-00578-f003:**
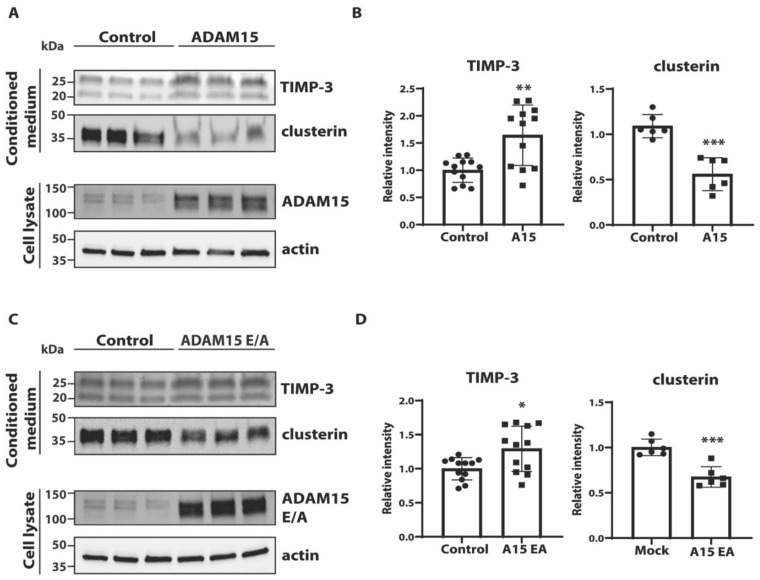
ADAM15 regulates levels of TIMP-3. (**A**) Levels of TIMP-3 and clusterin in the conditioned media and ADAM15 and actin in the cell lysates were evaluated by immunoblotting in ADAM15-overexpressing HTB94 cells and controls. (**B**) Densitometric quantifications of TIMP-3 and clusterin in the conditioned media of ADAM15 transfected and control cells are displayed as mean values ± standard deviation (* *p* < 0.05, ** *p* < 0.01, *** *p* < 0.005, Student’s *t*-test). (**C**) Similarly, levels of TIMP-3 and clusterin in the conditioned media, and ADAM15 and actin in the lysate of HTB94 cells transfected with ADAM15 E/A or controls were evaluated by immunoblotting and densitometric quantification (**D**).

## Data Availability

The data is contained within the article or Appendix A.

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
