# Peer review of "Quantitative Proteomics Reveals That ADAM15 Can Have Proteolytic-Independent Functions in the Steady State"

_membranes, 2022, doi:10.3390/membranes12060578_

Round 1
Reviewer 1 Report
This manuscript named ‘Quantitative proteomics reveals that ADAM15 can have proteo-2 lytic-independent functions in the steady state’ used unbiased proteomics to identify ADAM15 substrates and proteins regulated by the 25 proteinase in chondrocyte-like HTB94 cells. The authors used an array of proteomics methods to identify proteins regulated by ADAM15 and this work provides new insights on the biology of this proteinase and indicates that it can regulate a levels of other proteins involved in cartilage homeostasis in a catalytic-independent mechanism. This work would be of great interest and impact to the readership of Membrane. There remain questions regarding the experimental design. Those questions should be addressed before the manuscript is suitable for publication.
- The label-free methods might rely on the sample preparation and instrument parameters or performance, as well as the sequence coverage and the degree of complex sample fractionations prior to analysis in a mass spectrometer. It would be more convincing if the authors could combine this method with another quantitative proteomics method such as iTRAQ-based proteomics, TMT-based proteomics, SILAC-based proteomics, or absolute quantification.
Reviewer 2 Report
In this manuscript, Yang et al. performed proteomic analysis of secreted proteins to identify substrates and proteins regulated by A disintegrin and metalloproteinase 15 (ADAM15), a member of the ADAM family of 20 sheddases. They reported that ADAM15 silencing did not induce major changes in the secretome composition of HTB94 cells. However, overexpression of ADAM15 altered many secreted proteins compared to GFP-overexpressing controls. But the analysis did not identify any potential transmembrane proteins released by ADAM15 in the extracellular milieu. TIMP-3 expression was up due to ADAM15 overexpression, a major regulator of extracellular matrix turnover.
Despite their failure to identify any ADAM15 substrates, this is a well-designed and interesting study and I think this study moves the field of metalloproteinase forward and lays foundation for future studies.
Authors argued several possibilities for their failure to identify any substrates. One possibility they argued that proteins shed to the medium are already cleaved by endogenous proteases and further digestion with trypsin might make them undetectable. However, I disagree this possibility because there are many examples whether many metalloproteinases and their substrates were successfully detected by bottom-up proteomics. This is more like a comment rather than a criticism. If this is true, author can easily filter those cleaved substrates and analyze them directly by LC-MS as modern mass spec can identify cleaved proteins. They can also use partial digestion (quick trypsin digestion rather than standard 16h digestion). Again, this is simply a comment, and emphasizes that there are different ways to address this concern.
Round 2
Reviewer 1 Report
the authors have replied my questions and I am supportive of publication now.